# Soil Loss Estimation Using Remote Sensing and RUSLE Model in Koromi-Federe Catchment Area of Jos-East LGA, Plateau State, Nigeria

Andrew Ayangeaor Ugese [1], Jesugbemi Olaoye Ajiboye [1], Esther Shupel Ibrahim [2,3,4,*], Efron Nduke Gajere [2], Atang Itse [2] and Halilu Ahmad Shaba [5]

1   Department of Geomatic Engineering, Zonguldak Bülent Ecevit Üniversitesi, İncivez, Üniversite Cd., 67100 Zonguldak, Turkey
2   National Centre for Remote Sensing, Jos, Rizek Village Jos Eat LGA, Jos PMB 2136, Plateau State, Nigeria
3   Leibniz Centre for Agricultural Landscape Research (ZALF), Eberswalder Straße 84, 15374 Müncheberg, Germany
4   Earth Observation Lab, Geography Institute, Faculty of Mathematics and Natural Sciences, Humboldt-Universität zu Berlin, Unter den Linden 6, 10099 Berlin, Germany
5   National Space Research and Development Agency, Federal Ministry of Science and Technology, Abuja PMB 437, Nigeria
*   Correspondence: esther.shupel.ibrahim@hu-berlin.de

**Abstract:** Soil loss caused by erosion has destroyed landscapes, as well as depositing sterile material on fertile lands and rivers, clogged waterways and accelerated flash floods, declined the populations of fish and other species, and diminish soil fertility. In some places, erosion has also destroyed buildings, caused mudflow, create new landforms, displaced people, and slowed down the economy of the affected community by destroying roads and homes. Erosion is aggravated by climate change and anthropogenic factors such as deforestation, overgrazing, inappropriate methods of tillage, and unsustainable agricultural practices. In this study, remote sensing (RS) and geographic information (GIS) data and tools were used to model erosion and estimate soil loss in the catchment area of Koromi-Federe in Jos East, Plateau State Nigeria which is our study area. Soil loss estimation was performed using the revised universal soil loss equation (RUSLE) model and was computed by substituting the corresponding values of each factor inherent in the equation (rainfall erosivity, soil erodibility, slope steepness and slope length, cover management, and conservation practices) using RS and GIS tools. Soil data was obtained from the study area and analyzed in the laboratory, rainfall data, land cover, digital elevation model (DEM), as well as the management practice of the study area were the parameters computed in spatial analyst tool using map algebra based on RUSLE. The soil loss generated was classified into four classes and the results revealed 95.27% of the catchment with a tolerable loss of less than $10\ \mathrm{t/h^{-1}/y^{-1}}$. At 3.6%, a low or minimal loss of $10\text{–}20\ \mathrm{t/h^{-1}/y^{-1}}$, at 1.03% there exists a moderate loss of $20\text{–}50\ \mathrm{t/h^{-1}/y^{-1}}$, while there was and critical or high loss of $>50\ \mathrm{t/h^{-1}/y^{-1}}$ at 0.12% of the catchment. The result showed that critical soil loss in the catchment area is exacerbated by the influence of the slope length and steepness, and the amount of rainfall received. This poses great concern with annual rainfall projected to increase up to 12% in West Africa. However, our sensitivity analysis revealed that it can be reduced with the effect of vegetated cover and management practices. This is an important finding as it can guide sustainability practices to control erosion and the loss of valuable lands in the region, especially now under climate change.

**Keywords:** erosion; climate change; geospatial; GIS; rainfall; factors; management

## 1. Introduction

Soil is considered the Earth's fragile skin that anchors all life on Earth [1]. It is comprised of countless species that create a dynamic and complex ecosystem and is among

the most valuable resources, as both plants, animals, and people depend on soil for food and general survival. Just as the importance of soil cannot be overemphasized, its vulnerability to environmental degradation cannot be overlooked, among which is soil erosion. Thus, soil erosion is a major threat to biodiversity, as it affects crucial aspects of human, animal, and plant lives [2].

Soil erosion is the washing or movement of the top soil by erosion agents of water and wind [3]. It involves three processes of detachment, transportation, and deposition [4,5], and the major effect of soil erosion is soil and nutrients loss [6,7]. Although, soil erosion is one of the natural processes of soil formation which is influenced by topography, rainfall intensity, temperature, land use, and soil characteristics [8]. Soil erosion allows for the formation of soil, which is very important for sustainable agriculture as well as other soil-related activities, however, its resultant implication of excessive soil loss poses a great challenge to man.

Soil loss due to erosion is a source of concern when it is exacerbated by climate change and anthropogenic factors, such as deforestation, overgrazing, inappropriate methods of tillage, and unsustainable agricultural practices [9,10]. The effects of soil erosion go beyond the loss of fertile land [11], it has led to increased pollution and sedimentation in streams and rivers, clogging these waterways and causing a decline in the populations of fish and other species, while also increasing the risk of flash floods [12,13]. In another dimension, degraded lands are often less able to hold water and nutrients, which can aggravate erosion [14–16]. Sustainable land uses can help reduce the impacts of agricultural practices and prevent soil erosion and the loss of valuable lands. Soil erosion and soil loss have been a major ecological concern for environmentalist/conservators, farmers, and the government as well as the entire global community whose livelihood depend on soils, as thousands to millions of tons of soil are lost annually to erosion. On a global scale, Eswaran et al. [16] estimated the annual loss of 75 billion tons of soil costing the world about USD 400 billion per year, approximately USD 70 per person per year. The concerns are more critical in Africa as a result of climate change. Annual rainfall is projected to increase in West and East Africa from 0% to +12% depending on Representative Concentration Pathway (RCP) [17], local efforts exist to understand resident climate change in Nigeria [18,19]. As such, the government of Nigeria through the World Bank has set up the Nigerian Erosion and Watershed Management project (NEWMAP) worth USD 500 million to combat or reduce erosion related problems in the country [19]. There is first the need for robust tool in assessing spatial distribution of soil loss to tackle the menace head on.

Remote sensing and Geographic Information System (GIS) tools have become robust tools in estimating soil loss and monitoring and managing erosion. The always-growing availability of earth observation data and the well-established use of Geographic Information Systems (GIS) lead to the development of automated geospatial toolboxes for the estimation of soil erosion [20,21]. These tools have been used in different parts of the world to predict erosion and soil loss; in South Africa [22], in Germany [23] in Italy [24], in India [25,26], in China [27], in Ethiopia [28], in different parts of Nigeria [29–31], and so on. In the study of soil loss and in the quest of estimating the amount of soil loss in catchment areas, there has been the development of tools and methodologies by researchers. The choice of these methodologies often depends on data availability, the complexity of the study areas, technical know-how, and the process involved in the application [32]. Some of the most commonly employed models in soil loss estimation are the Universal Soil Loss Equation (USLE) [33], its revised version, the revised universal soil loss equation (RULSE) [34], Water Erosion Prediction Project (WEPP) [35,36], Pacific South-West Interagency Committee (PSIAC) [37], the Modified-PSIAC for arid and semi-arid areas [37,38], the Erosion Potential method (EPM) [39], ICONA [40] among others.

While their choice clings to various factors [32], Amiri [41] uses the EPM in estimating soil erosion and sedimentation yield in the Ghareh Aghach basin in Iran. The EPM estimates soil loss using a variety of environmental factors, however, Rafahi [42] opined its accuracy in estimating yield is less than other known models. Bayramin [40] employed the ICONA

in the soil erosion risk assessment of the Beypazari area. The ICONA model uses the slope and the geological map (lithofacies layers) to produce soil erodibility, which is referred to as the potential erosion risk map (PER). Hereafter, it included soil protection layers or parameters derived from the vegetation cover (NDVI) and the land use type to estimate soil loss. The PSIAC/MPSIAC has also been used and prioritized for their reliability, capability, accuracy, and precision in estimating soil loss in arid and semi-arid regions, and also for all kinds of erosion including gully erosion [37,38,43,44].

However, the most implemented universally accepted model used for estimating soil loss is the universal soil loss equation (USLE) [33], and its revised version (RUSLE) [34] is the most commonly used model for estimating long-term average soil loss, especially with remote sensing data and tools [24,45,46]. The RUSLE model is adaptable and applicable in diverse scales [46], including Nigeria [29–31]. In the Nigeria context, the RUSLE is often employed in soil loss estimation [47–51]. Obiahu and Elias [47] deployed RS data and the RUSLE in assessing the effect of LULC on the rate of soil erosion in the catchment of Afikpo North areas over a period of 20 years (1996–2016) with a result citing an increase in food demand and mining activities as the driving force for the LULC changes which influences the soil erosion rate between the period. Similarly,, Adediji et al. [30] using RS and GIS data and also adopted the RUSLE in soil loss estimation in Katsina with findings identifying the slope factor to be of major significance exacerbating soil loss. Emeriobeole [51] also explored the RUSLE in soil loss estimation of Imo, in Imo state, Nigeria. In the case of Emeriobeole [51], deforestation and land clearing for agricultural purpose, urban development and the slope steepness are identified as the cause for wide spread erosion in the catchment of Imo. Okenmuo [49] employed RUSLE in estimating soil water erosion in Obibia river watershed of Anambra with a result validating the RUSLE as an accurate and cost-effective means for soil erosion prediction. Dike [50] also applied the RUSLE in their estimation of soil loss rate in Urualla, Nigeria. The above listed studies reveals the frequency in the use of RUSLE due to its simplicity and suitability for integration in GIS. As such, in this study, on the basis data accessibility, nature of terrain, the type of erosion experienced in the study area, and the wide acceptance/implementation of RUSLE across various watersheds in Nigeria, we tested the RUSLE model to estimate soil loss in a catchment in Plateau state, Nigeria where erosion has increased in the recent past and robust scientific understanding using remote sensing and GIS data and tools are currently lacking.

## 2. Materials and Methods

### 2.1. Study Area

The Koromi-Federe catchment area is situated in Jos-East Local Government of Plateau state and is located between 90°0′01″ E, 90°49′38″ N and 9°10′15″ E, 100°00′00″ N (Figure 1). The Kurumi catchment covers an area of 222.133 km². According to the Köppenen climates classification, the study area is located within equatorial (AW). Elevation is high, and situated within the boundary of the Maijuju rock formation. Temperature is generally low as an element of elevation, with an average temperature that ranges between 17 °C and 25 °C [52]. The mean monthly rainfall is 180 mm, with a peak of >228 mm between June and August. A recent increase in precipitation is recorded in the region under climate change, and more serious erosion cases are recorded (field observations; see samples in Figure 1a,b). This presents concerns as the Koromi catchment is an important economic hub in terms of agriculture for both animal and crop production. Agriculture is the main activity of the people, growing majorly cereal such as maize, millet, and rice, and important vegetables such as pepper, cucumber, cabbage, okra, peas, and green beans in both rainy and dry seasons [53]. The area is green with the availability of grasses, shrubs, trees, and streams (Figure 2), thereby making it suitable for cattle owners to graze for the production of milk and meat. As such, large camps of grazing activities are dominant in the region. With elevations ranging between 817 m to 1761 m, the terrain is highly undulating with mountainous and plain landscapes which makes it highly susceptible to erosion, more

so that the soil is sandy Loam which is generally prone to erosion (field survey, 2015). The tributaries in the catchment (Figure 1) are the source of the flow of many rivers in northern Nigeria including the Gongola, Hadejia, and Yobe rivers (www.nigeriagalleria. com; accessed on 30 March 2019). However, erosion has recently increased in the region causing devastating effects on farmlands, infrastructures, housing, and so on, as shown in Figure 1a,b.

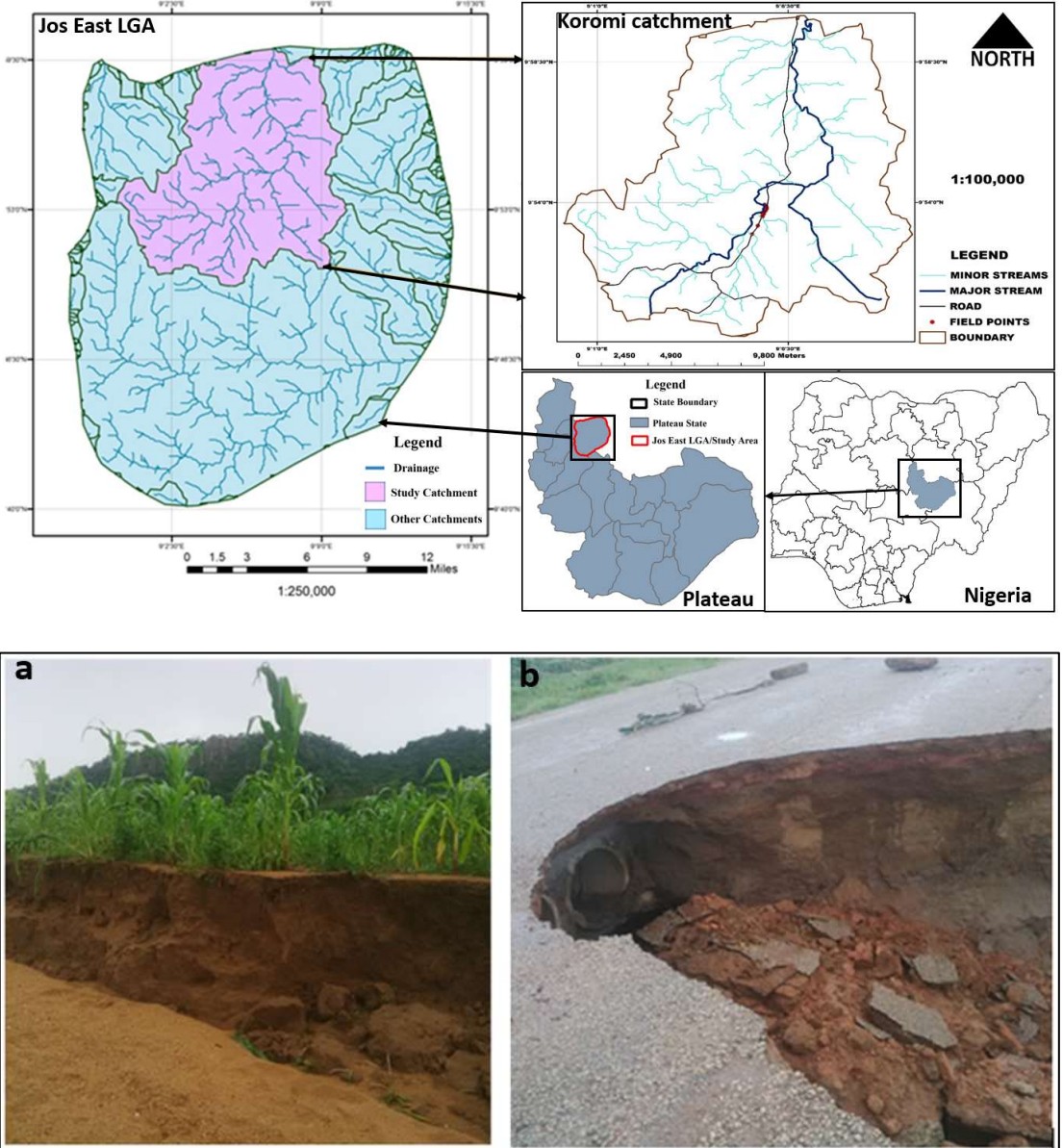

**Figure 1.** Map of Jos East LGA showing the location of Koromi catchment and drainages. (**a**). is a cultivated field in the Koromi catchment partially eroded; (**b**). is the main Fobur to Ferere road eroded. Both pictures were captured in 2015 during the field work exercise.

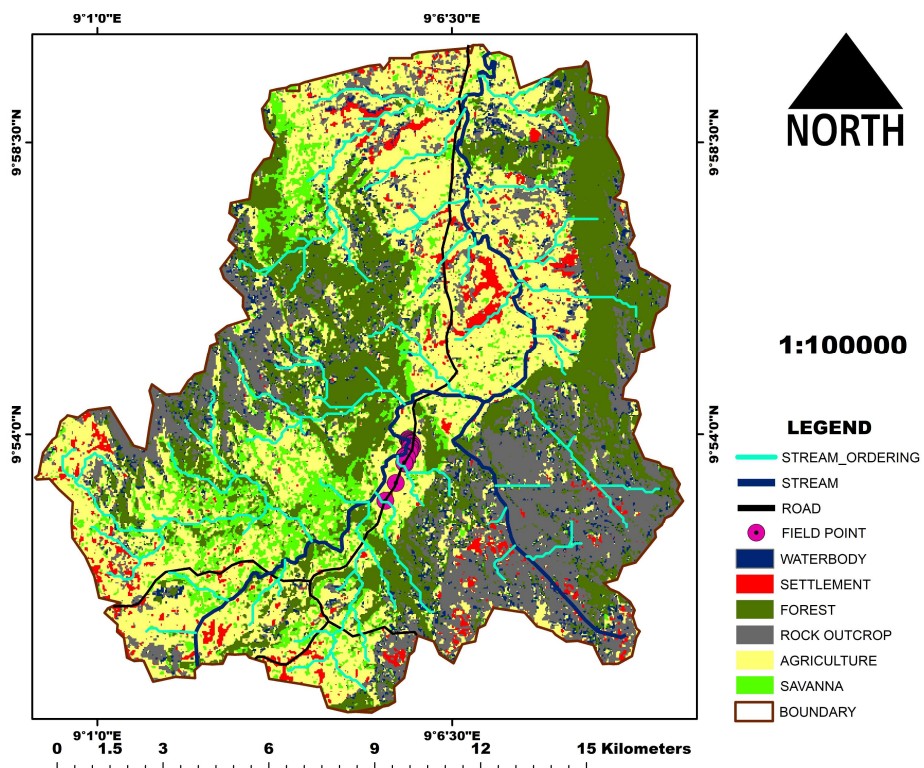

**Figure 2.** The land cover of Koromi catchment.

*2.2. Description of the RUSLE Method*

The methodology was based on the principle and formulas of the RUSLE [34], and the analysis was carried out using "map calculator" of the ArcGIS "map algebra".

The RUSLE mathematical equation is expressed below:

$$A = R \times K \times LS \times C \times P \tag{1}$$

where:

$A$ is the average annual soil loss per hectare (t/ha$^{-1}$/y$^{-1}$),

$R$ is the rainfall run-off erosivity factor (MJ mm/ha$^{-1}$/h$^{-1}$/y$^{-1}$)

$K$ soil erodibity factor (t ha/h/ha$^{-1}$/MJ$^{-1}$ mm$^{-1}$)

$LS$ is the slope-steepness factor (dimensionless)

$C$ is the cover management factor (dimensionless)

$P$ is the conservation practices (dimensionless)

2.2.1. Rainfall Erosivity Factor (*R* Factor)

Rainfall erosivity is the first factor required in the equation. The *R* factor is based on rainfall impact in the form of kinetic energy, and it also projects the rate and quantity of run-off which is directly interconnected with a particular precipitation event [33]. According to Wischmeier and Smith [33], a period of 20–25 years is recommended for computing the average annual rainfall. As such, the monthly rainfall for the study area was collected from five meteorological stations near the study area for a period of 30 years, collected by the Nigerian Meteorological Agency (NIMET).

The monthly averages for the 30-year period were used to calculate rainfall erosivity using the mean annual rainfall in accordance with the following formula:

$$R = 8.12 + 0.562P \tag{2}$$

where *R* is the rainfall erosivity factor and *P* is the mean annual rainfall (mm).

The data were then interpolated in the ArcGIS to produce continuous rainfall data for each grid cell using the spatial analyst tool.

### 2.2.2. Soil Erodibility (*K* Factor)

The *K* factor (soil erodibility factor) is next in the equation. This *K* factor is the estimation of the resistance of soil against erosion due to the impact of raindrops (interception) and the rate and amount of run-off produced for that rainfall impact, under a standard condition usually depending on geological and soil features [54]. The *K* factor was generated from the results of our analyzed soil samples. For the purpose of this study, soil texture, structure class, organic matter, and permeability class were considered. Using their corresponding values, it was calculated using the following formula from the 21 samples we collected on the field from July to August 2015. Soil tests were carried out in the soil department of the Federal College of Forestry, Jos Plateau State Nigeria. The interpolation of the soil samples was carried out in ArcGIS spatial analyst tool to generate continuous soil data. The *K* factor was calculated using the formula below:

$$K = 2.8 \times 10^{-7} \times (12 - OM) \times M^{1.14} + 4.3 \times 10^{-3} \times (s - 2) + 3.3 \times 10^{-3} \times (P - 3) \quad (3)$$

where:

*K* = soil erodibility factor.
*OM* = organic matter content
*P* = soil permeability class
*S* = soil structure
*M* = particle size parameter (%silt + % very fine sand) $\times$ (100 − %clay)

### 2.2.3. Slope Length and Slope Steepness (*LS* Factor)

The *LS* factor is a combination of two topographic factors, which are: slope length (*L*) and slope steepness (*S*). Usually, as the length of the slope increases, the amount and rate of cumulative run-off also increase. Likewise, as the land slope increases, the run-off velocity also increases and results in a massive erosion incidence [46]. Digital elevation model (DEM) was extracted from the Shuttle Radar Topographic Mission (SRTM) at a 30 m resolution and was downloaded from the United States Geological Survey department (accessed from; https://earthexplorer.usgs.gov/; accessed on 15 February 2016). The DEM was used to generate slope length and slope steepness, also using the spatial analyst tool in ArcGIS, where slope and flow accumulation were generated. These were then multiplied in the map algebra in the spatial analyst tool to generate the *LS* factor.

The following formula was used:

$$LS = POW\left(FA \times \frac{cs}{22.13}\right) m \times \left(0.065 = 0.045.s + 0.0065.s^2\right) \quad (4)$$

where

*FA* = Flow Accumulation
*m* = slope value
*s* = slope DEM
*cs* = cell size

### 2.2.4. Cover Management Practices (*C* Factor)

Next in the RUSLE equation is generating the *C* factor. The *C* factor accounts for how croplands and crop management causes soil loss varying from soil losses occurring in bare or fallow areas [55]. The land cover map of the study area was generated using a Landsat 8 satellite image at a 30 m spatial resolution (with date, raw, and path information: LANDSAT/LC08/C01/T1_TOA/LC08_188053_20151016). We performed image pre-processing and corrections (atmospheric and radiometric corrections). This image is 100% cloud-free and dated 16 October 2015. This date captures more accurately the day of

year when major land cover and land uses mapped are present. Crops are dominantly still on the field and not harvested, this window is the peak of greening in woody vegetation, both seasonal and perennial rivers and streams have water, and any bare areas within this window are dominantly bare areas not due to seasonal changes. A false color band combination (bands band Band-5 (NIR), Band-4 (red), and Band-3 (green)) was used to generate an image composite. The data was also downloaded from the United States Geological Survey department (accessed from; https://earthexplorer.usgs.gov/; accessed on 15 February 2016). Supervised image classifications using maximum likelihood algorithm in Erdas imagine software was carried out to generate the land use/land cover [53,56]. The validation of the land cover was achieved using 60 land cover field samples collected in 2015, and as carried out using cross-validation approach [57] and an overall accuracy of 78% was achieved. The land cover classes generated are water, agriculture, rock outcrop, Savanna, settlement, and forest (Figure 2). The same Landsat image was used to generate a normalized difference vegetation index (NDVI). NDVI is a measure of the greenness or vigor of vegetation [58] and was generated (map is in Figure 3), using band 4 (red) and band 5 (near-infrared band) and this formula;

$$\text{NDVI} = N\text{IR} - \text{Red}/N\text{IR} + \text{Red} \tag{5}$$

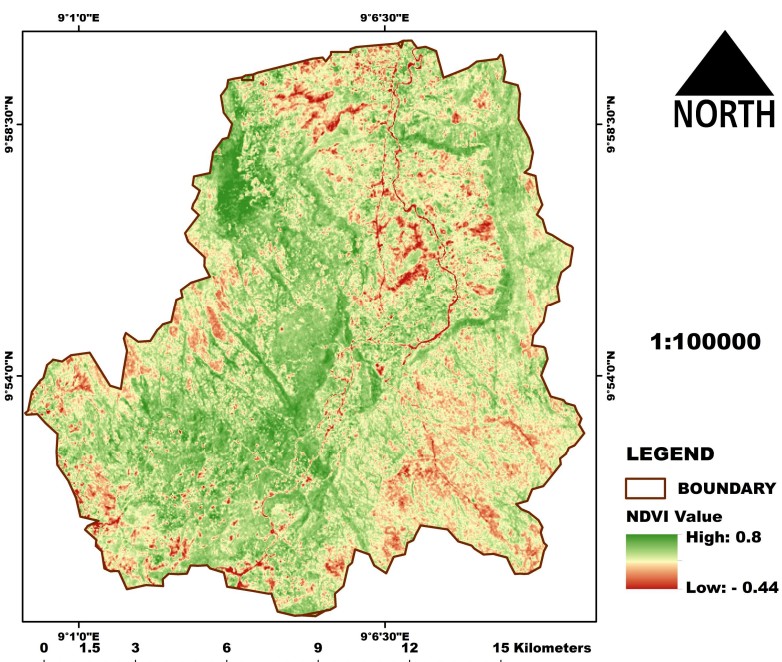

**Figure 3.** Normalized difference vegetation index (NDVI) of Koromi catchment.

The expression below was used to estimate the *C* factor

$$C = exp\left[-a\frac{NDVI}{(\beta - NDVI)}\right] \tag{6}$$

where alpha = 2 and Beta = 1

### 2.2.5. Management Factor (*P* Factor)

The last factor is management (*P* factor). This is the ratio of soil loss using a specific support practice to the corresponding loss with upslope and down-slope tillage [34]. The *P* factor map was derived from the land cover and weighed values assigned to the land uses from 0 to 1, in which the highest value is assigned to areas with no conservation practices (open areas, water, bare and grasslands), which are dominantly areas with conservation

practices, while the farmlands (agriculture area) are assigned different thresholds below 1 based on slope percentage as recommended by Wischmeier and Smith [33].

After the generation of the *R*, *C*, *P*, *LS*, and *K* factor maps, annual soil loss (*A*) was estimated using the mathematical equation (Equation (1)) described above. However, next, we tested the model sensitivity based on field data and factor maps generated. The entire methodological workflow for generating soil loss is summarized in Figure 4 below and data summary and sources are in Table 1.

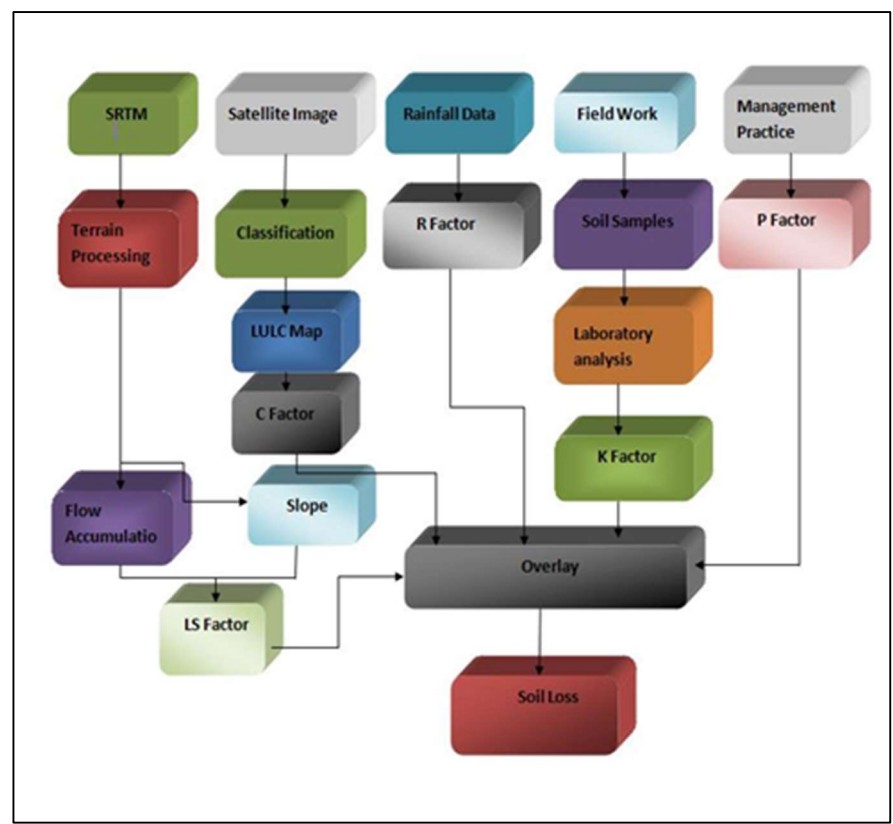

**Figure 4.** Illustration of workflow.

**Table 1.** Data summary.

| Data | Type | Spatial Resolution | Source |
|---|---|---|---|
| Rainfall | Vector | - | NIMET |
| Soil | Vector | - | Field work |
| Elevation and slope (derived from SRTM) | Raster | 30 m | https://earthexplorer.usgs.gov/ accessed on 15 February 2016 |
| Land cover and NDVI (derived from Landsat 8, dated 16 October 2015). | Raster | 30 m | https://earthexplorer.usgs.gov/ accessed on 15 February 2016 |

### 3. Results

*3.1. Rainfall Erosivity Factor (R Factor)*

The *R* factor is the erosive potential of rainfall using 30 years of rainfall records. Rainfall erosivity is greatly influenced by the volume, intensity, duration, and pattern of rainfall. Our results revealed that the average annual *R* factor values ranges from 42.9 in the southwest to 44.3 MJ mm/ha$^{-1}$/y$^{-1}$ in the northeastern parts of the study area (Figure 5).

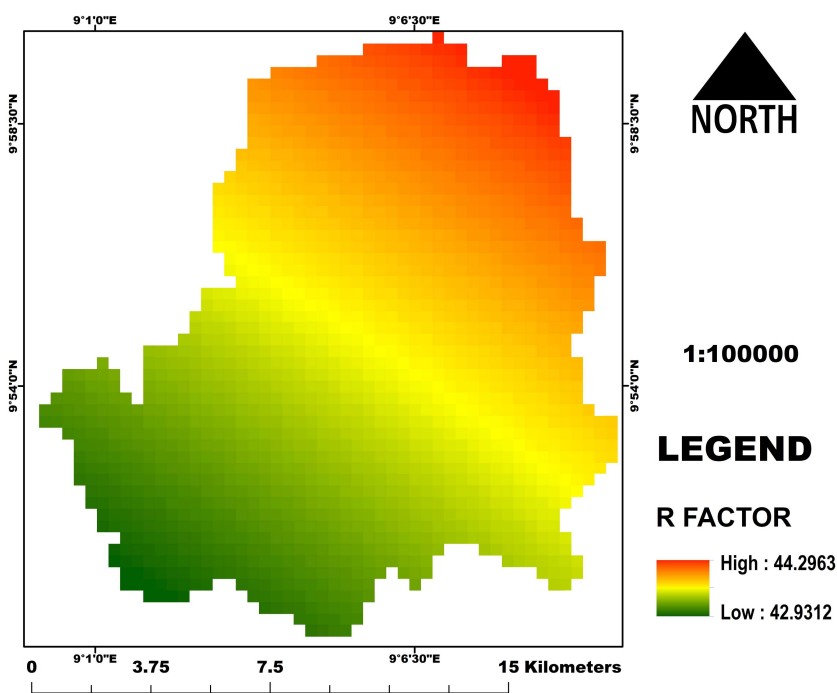

**Figure 5.** The *R* factor map.

### 3.2. Soil Erodibility Factor (K Factor)

The *K* factor represents the resistance or erodibility of the soil under varying conditions and land cover. Erodibilty depends essentially on organic matter, structure, texture, and permeability [59]. In the study area, the major soil texture classes found were sandy loam, sandy clay, loam, loamy sand, and sand with sandy loam the dominant which is highly susceptible to erosion and could explain the high degree of erosion of the area (Figures 6 and 7). The soil analysis also revealed very high sand content in the soil samples (Figure 8), further implying susceptibility to erodibility.

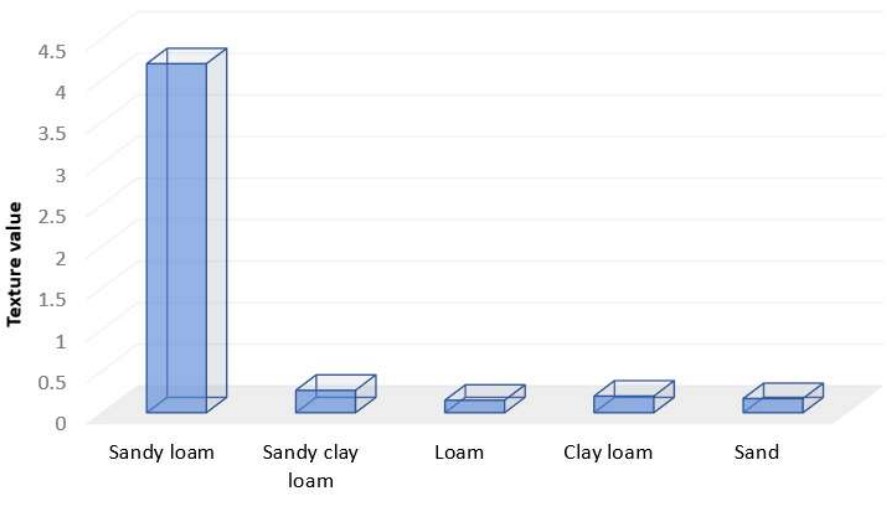

**Figure 6.** The distribution of soil texture in Koromi catchment of Jos East LGA.

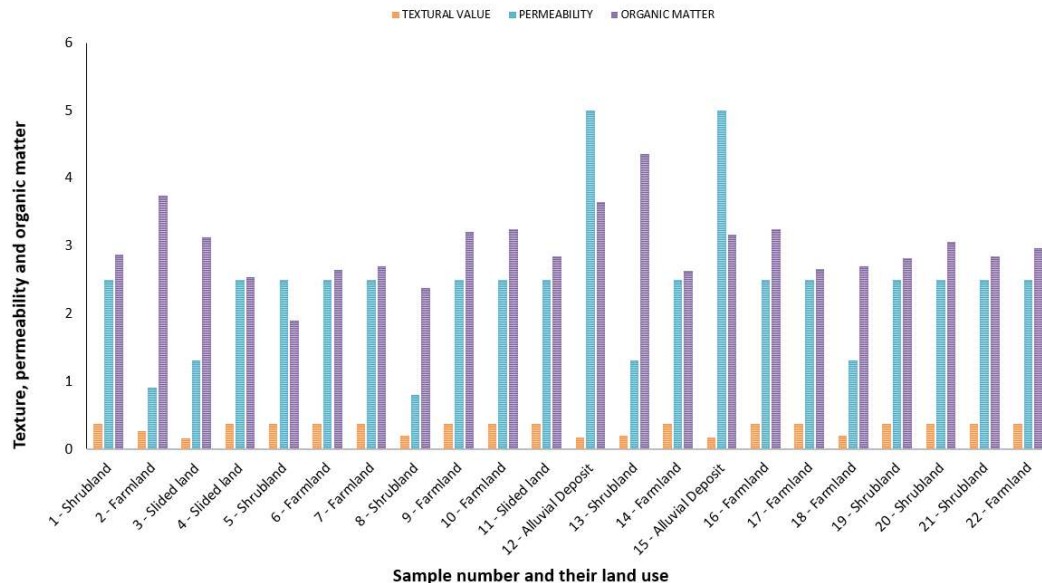

**Figure 7.** The distribution of soil characteristics within different land cover types in Koromi catchment of Jos East LGA.

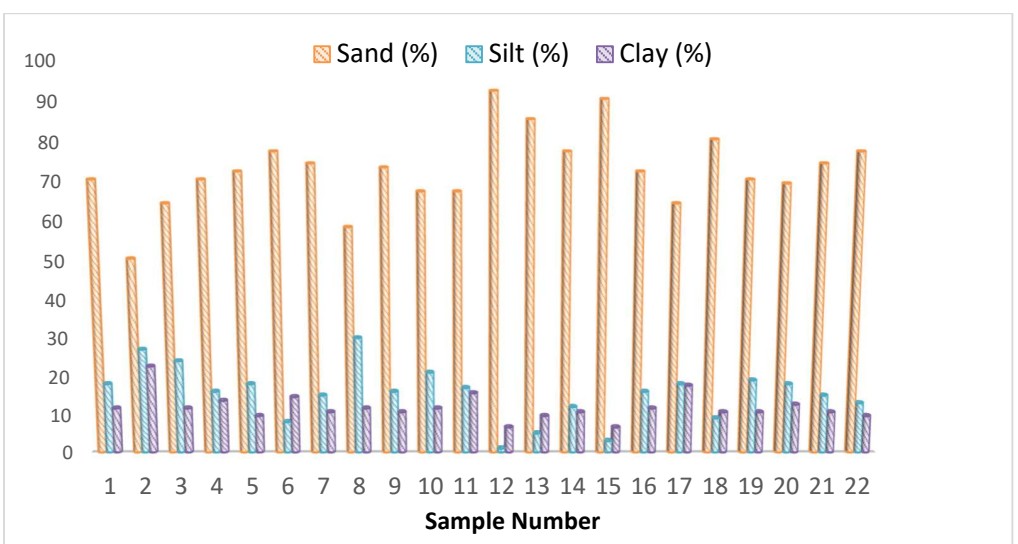

**Figure 8.** The distribution of soil content within different soil samples analyzed in Koromi catchment of Jos East LGA.

The result of the *K* factor revealed soil erodibility susceptibility from 0.216 to 0.245. The results implies that, the higher the erodibility factor, the greater the inherent potential to erode. The northern parts of the region are therefore more susceptible to erosion based on Figure 9. The difference between the highest and lowest *K* factor in the region is however, narrow (0.028), as soil types in the region are similar.

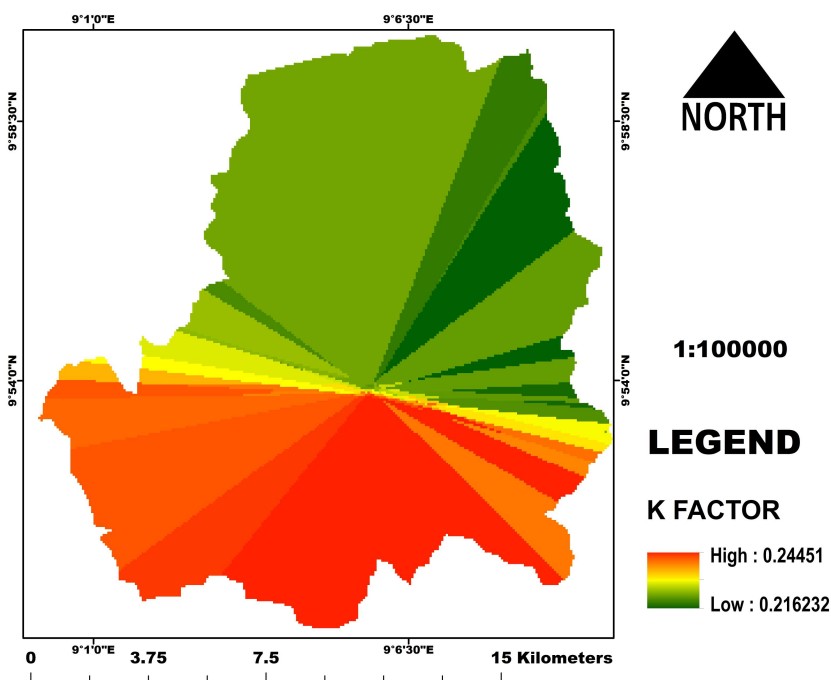

**Figure 9.** The *K* factor map.

### 3.3. Slope Length and Slope Steepness (LS Factor)

The *LS* map describes the impact of topography on soil erosion using slope length and steepness. The shorter the slope length the steeper the slope, hence greater cumulative runoff. The highest point from the *LS* factor is 45.4648 (Figure 10) and these areas correspond with the locations of the high elevations and very steep slopes in the study area.

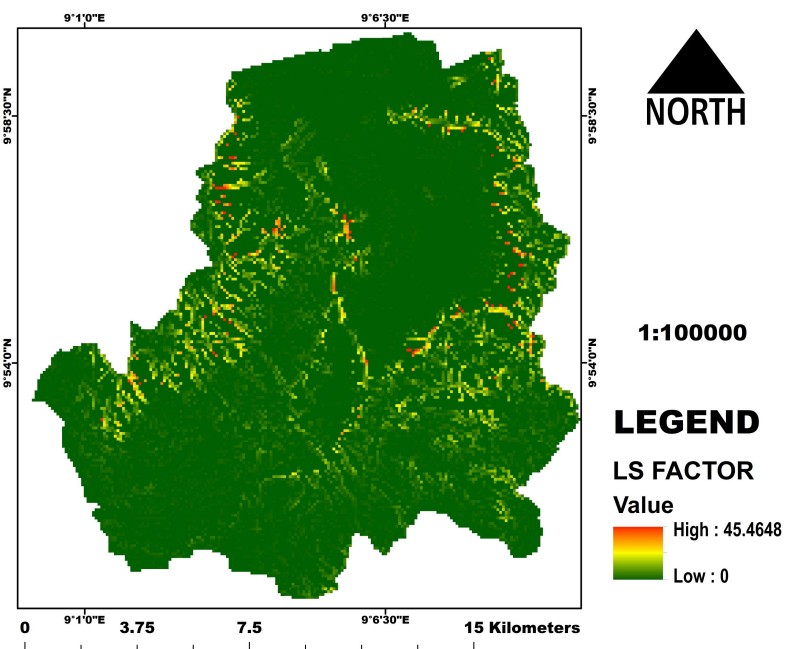

**Figure 10.** The *LS* factor map.

### 3.4. Crop Management Factor (C Factor)

The *C* factor map is the relative effectiveness of soil and crop management systems in terms of preventing or reducing soil loss. This indicates how conservation plans will affect the average annual soil loss. It shows the ratio of soil loss from land cover un-

der specified conditions to that from continuously fallow and tilled lands. The *C* map (Figure 11) revealed higher values of 1.28 in the region and correspond with bare, cultivated, water, and built areas, while low values correspond with the forest and savanna land cover (Figures 2 and 11).

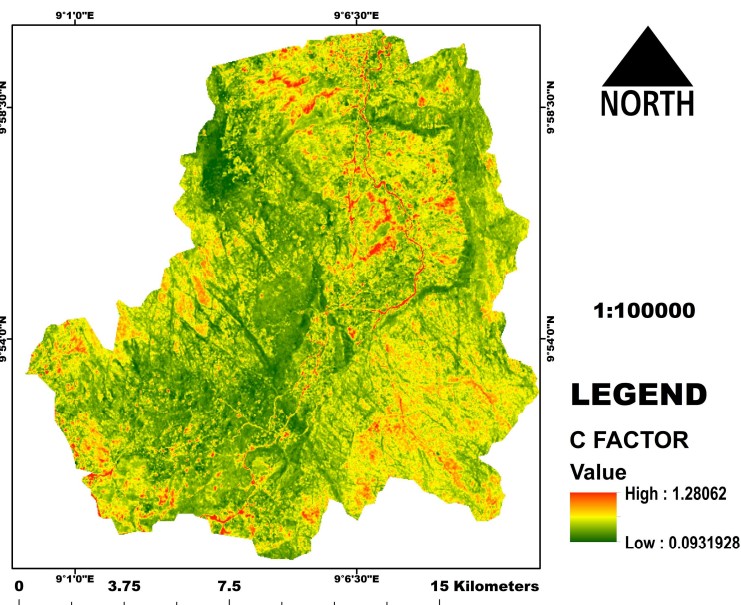

**Figure 11.** The *C* factor map.

*3.5. The Soil Management Practice (P Factor)*

The *p* value reflects the effects of practices that can potentially reduce or increase the amount and rate of water runoff and thus reduce or increase the rate and degree of erosion. Common support practices include cross-slope cultivation, strip cropping, contour farming, terracing, grassed waterways, etc. The *P* factor map shows areas with no conservation practices as 1 while areas with relative conservation practices as 0.2 (Figure 12). The results depict that most parts of the area are highly vulnerable based on this factor (Figure 12) as most farming activities in the area are carried out with no or little consideration of technical conservation practice.

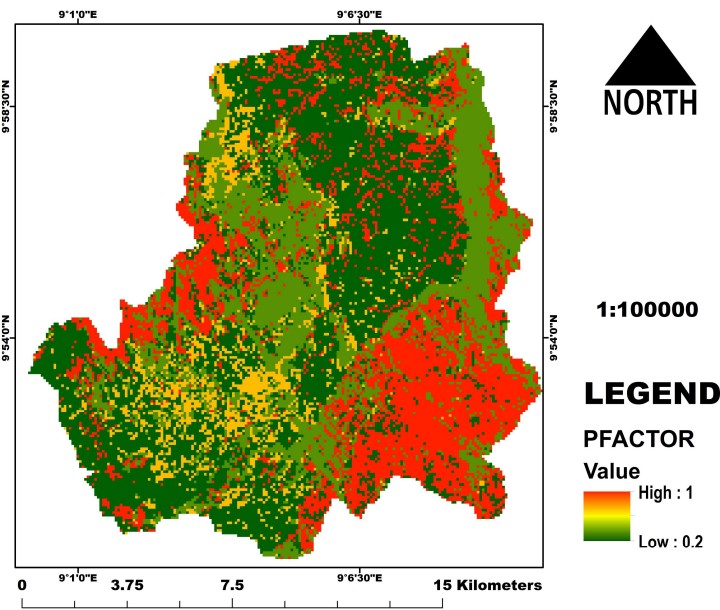

**Figure 12.** The *P* factor map.

### 3.6. Soil Loss Analysis

The soil loss estimation using the RUSLE model was computed by multiplying each factor as highlighted. However, two parameters were considered very sensitive in the model. This is based on our model calibration analysis. This is what we termed "soil erosion susceptibility", which is the likelihood of soil loss if no crop management or erosion control practices are put in place. Therefore, it considers *K*, *R*, and *LS* factor only. This susceptible map revealed that the study area falls within a low to medium susceptibility based on the combination of the factor maps, but accounting for up to 441 t/h$^{-1}$/y$^{-1}$ in the high areas (Figure 13).

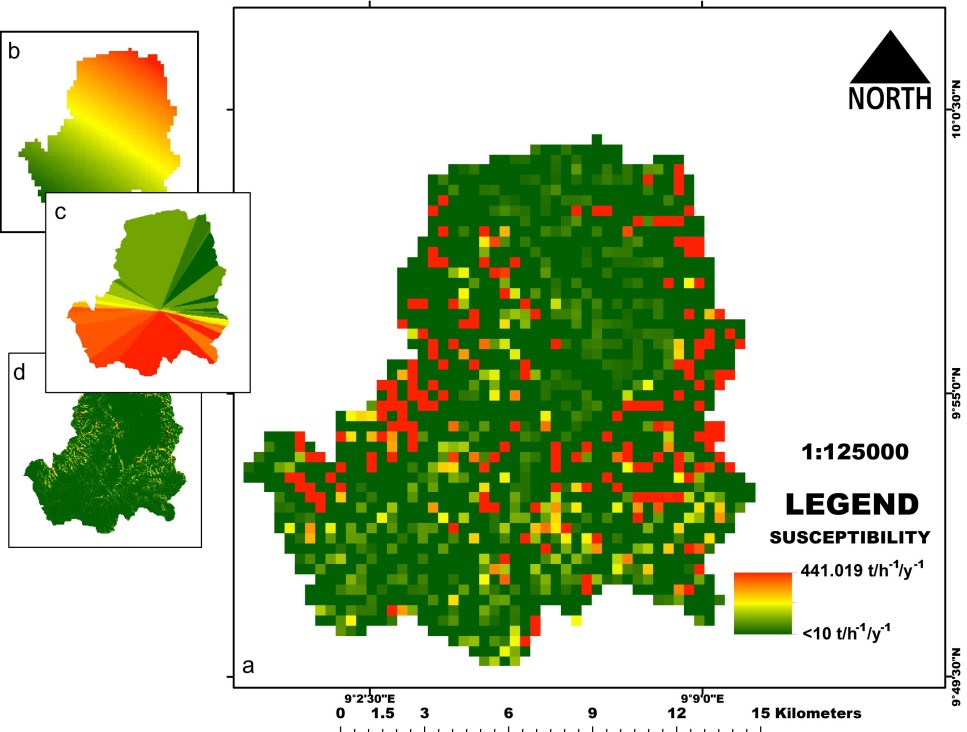

**Figure 13.** (**a**) The erosion susceptibility map; (**b**) *R* factor; (**c**) *K* factor; (**d**) *LS* factor.

### 3.7. Soil Loss Hazard

This final map in Figure 14 is the product revealing soil loss from the addition of all the factor maps, including conservation which we termed "soil loss hazard". Comparing the soil loss hazard with the soil erosion susceptibility map (Figure 13), a contrasting map is revealed in Figure 14. In the susceptibility map (Figure 13), annual soil loss was up to 441 t/h$^{-1}$/y$^{-1}$, while in the annual soil loss hazard map the highest value of 84 t/h$^{-1}$/y$^{-1}$ is revealed. Implying a significant decrease in potential annual soil loss if *C* and *P* (conservation and management) factors are introduced.

For a better understanding of the final result, the soil loss layer was classified into four classes (Table 2). The summary shows that about 95.27% of the catchment with a tolerable loss of less than 10 t/h$^{-1}$/y$^{-1}$, a low or minimal loss of 10–20 t/h$^{-1}$/y$^{-1}$ been 3.58% of the Koromi-Federe catchment, 1.03% which amount to 20–50 t/h$^{-1}$/y$^{-1}$ of a moderate loss and critical or high loss of >50 t/h$^{-1}$/y$^{-1}$ (0.12) of the catchment.

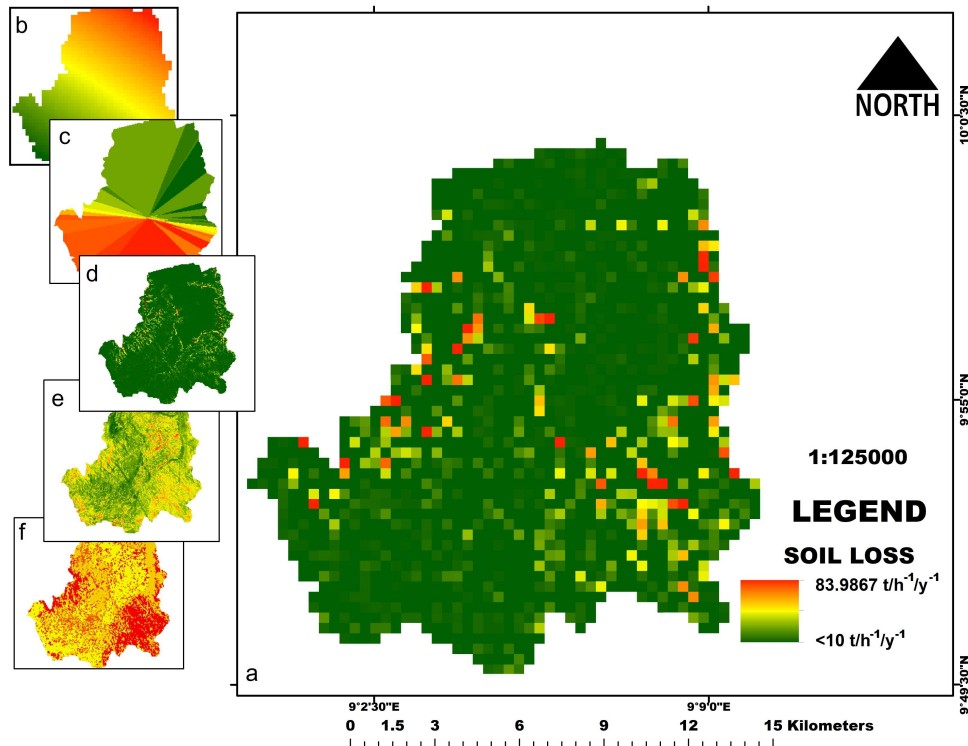

**Figure 14.** (**a**) The soil loss map; (**b**) *R* factor; (**c**) *K* factor; (**d**) *LS* factor; (**e**) *C* factor; (**f**) *P* factor.

**Table 2.** An estimation of the annual soil loss rating.

| Rate of Loss | Soil Loss $(t/h^{-1}/y^{-1})$ | Percentage (%) |
|---|---|---|
| Tolerable/negligible | <10 | 95.27 |
| Low loss | 10–20 | 3.57 |
| Moderate loss | 20–50 | 1.03 |
| Critical loss | >50 | 0.12 |

## 4. Discussion

In the recent past, there has been an increase in the incidences of soil loss as a result of erosion in Nigeria [29–31,49–51]. Erosion is expected to intensify due to climate change [23], as rainfall patterns have intensified; its frequency has increased and is projected to be catastrophic, especially in West and East Africa [17]. It is therefore critical to estimate, monitor, and manage erosion repetitively and sustainably. RS and GIS are prominent tools for the estimation and management of soil loss in various spatial scales [46]. We likewise tested the capability of RS and GIS data and tools to estimate soil loss using the RUSLE model in parts of Nigeria.

Our sensitivity analysis revealed that up to 441 $t/ha^{-1}/y^{-1}$ of soil loss is possible if no cropping management or erosion prevention practices are put in place. However, when cropping management and erosion prevention practices were tested, the soil loss reduced to 84 $t/h^{-1}/y^{-1}$. This shows that soil management and conservation practices can potentially reduce the amount and rate of soil loss in the Koromi–Federe catchment. The *K* values between 0.21 and 0.24 $t/ha^{-1}/y^{-1}$ are predominantly within sandy loam soils and generally susceptible to erosion. Considerably, the *P* and *C* factors act as shield to soils, hence the more the conservation and management practices, the less the erosion. This is consistent with reports by Fashae et al. [31] for parts of Oyo State Nigeria, where they also reported land cover as the most important factor influencing erosion. Likewise, Rotschek et al. [23] reported a similar finding for Germany, implying that land use, soil management and soil properties have higher effects compared to changes in precipitation patterns. In general, soil loss studies have conveyed that, vegetation cover especially

along with slope length and steepness as a highly influential factor for managing soil loss [31,46,60]. This is owed to the fact that vegetation cover directly affects raindrops and the soil particle detachment, by dissipating raindrop energy before reaching the soil [46]. This significantly reduce soil detachments and subsequently erodibility.

Nonetheless, our sensitivity analysis revealed that soil erosion is highly influenced by *LS* and *R* factors. This is more obvious at the hilly and steep slopes of the catchment revealing high soil loss risk despite high vegetation cover. Surprisingly, the areas under high risks of erosion in both Figures 13 and 14 correspond with areas under forest cover (Figure 3) and high NDVI values (Figure 4). Implying that vegetation reduces the influence of soil loss to a minimum threshold (Figures 13 and 14), and not total eradication in terms of managing soil erosion dues to *LS* (Figure 10) in the Koromi catchment. Our findings are inconsistent with the findings by Adediji et al. [30] for Katsina state in Nigeria, where they reported *LS* as the most significant and sensitive factor in their study region. While, Gobin et al. [29] reported that gully erosion in parts of Nsuuka in Southwestern Nigeria is influenced by all factors (infrastructure, geohydrology, topography, vegetation, and land use) although they recorded higher soil loss on escarpment than on plateau soils. In another vein, the areas with extreme susceptibility and high annual soil loss in our results overlay with areas under high precipitation (Figure 5). This corresponds with reports by Langbein and Schumm [61] who quantified the non-linear relationship between precipitation and sediment yield and found that vegetation and precipitation exert competing effects, implying the force of intense precipitation can reduce the impact of vegetation cover. More recently, Srivastava et al. [62] likewise reported that despite high vegetation cover, there can be more runoff as a result of high precipitation. Therefore this implies greater erosion risk under climate change and increase in erosion incidences [23]. Even though reports exist for a forecasted decline in precipitating amount, intensity, and duration on the Jos Plateau [18], recent climate records for the region reveal a persistent increase in precipitation amount (https://weatherspark. com/; accessed on 25 September 2022), backed by observed changes in both rainfall amount and intensity. The Jos East area receives high rainfall, especially in the months of July and August. The Kurumi catchment is located vicinity of hills and forms tributaries, which are the origin of the many famous rivers in Northern Nigeria such as River Gongola, River Hadejia, and River Yobe. This area will continue to encounter devastating soil loss especially now with the increase in rainfall intensity and duration reported in the region under climate change.

In general, the devastation of erosion and soil loss in the Koromi catchment is accelerated by the hilly terrain in the area, the soil type (sandy loam) which is highly susceptible to erosion, and high rainfall factors acting on the catchment which is further aggravated by a lack of technical farming and conservation practices. Silt and sand content in the region is high (Figure 8), and Mhangara et al. [22] discussed that soils become highly erodible if the presence of silt percentage is high. Of importance to note is that the farmlands generally lie on the flood plain (minor stream orders) hence there is devastation as the runoff finds ways to escape from its obstructions which is largely the cultivated lands. This is can result in negative impacts on vegetation cover, soil productivity for agricultural activities, ground water contamination as also reported by Fashea et al. [31]. Additionally, the major road connecting Jos to the LGA headquarters was destroyed as at the time of this study in 2015 (see Figure 1), disconnecting the settlements around its vicinityas a result of erosion. It was concluded that an underestimation of the erosive power of the runoff from the major drainage channel during the road construction resulted in the destruction of the road. There is a need to factor this component in future constructions to select optimal reliable and sustainable paths for road constructions.

In terms of model (RUSLE) efficiency, Eisazadeh [43], on the basis of accuracy and precision, however questioned the reliability of RUSLE compared to the MPSIAC. Elisazadeh's [43] study in ten basin upstream reservoirs of West Azerbajan province of Iran answers the question after evaluating the two models (MPSIAC and RUSLE) in estimating soil loss and sediment yield. Their findings reflects the strength of MPSIAC over RUSLE.

It shows that MPSIAC has a high capability in soil erosion and sediment yield in basins upstream reservoir compared to the RUSLE with a higher bias rate. Questions are also directed at the ability of the RUSLE to accurately estimate soil loss in the case of gully erosion aside from the usual sheet and rill erosion it's often employed [38,43,47]. Fiçici [44] on the basis of accuracy identifies the MPSIAC to give more reliable data while comparing the MPSIAC and RUSLE methods in soil erosion analysis at the Madra Dam basin of Turkey. Other reasons identified for the superiority of MPSIAC over the RUSLE hinge on the number of parameters it weights to present a more precise soil loss estimate [38]. According to Daneshvar [38], the PSIAC or/and MPSIAC model employs nine factors which are, surface geology data, soil data, climate data, runoff, topography, vegetation cover, land use, upland erosion, and channel erosion data while RUSLE, on the other hand, adopts five (5) factors: rainfall erosivity, soil erodibility, slope factor, cover factor, and management practices [33]. Regardless of Elisazedeh's [43] prioritization of the MPSIAC over RUSLE, Danesh [38] reiterated that the MPSIAC method is specially designed for implementation in arid and semi-arid regions as simulated in the United States of America at its creation in 1982 after the earlier introduction of the PSIAC in 1968. Notwithstanding, RUSLE/USLE has also been suggested for use for their convenience and minimum data requirement, especially for data-scarce regions such as Africa. The RUSLE/USLE is also recommended for GIS compatibility on various scales, and mostly its application in terrain with soft undulating slope [48], as such widely used in the Nigerian context [29–31,47–51] and we consider its performance in our study region sufficient for soil loss estimation. However, we recommend further studies to adequately understand vegetation–precipitation–soil loss relationships and to identify effective management strategies, especially in hilly and steep slope terrains. We also recommend a deeper understanding of erosion in relation to different vegetation types, and likewise infiltration rate in future studies. This will direct pragmatic and sustainable management strategies for regulating erosion in Nigeria.

## 5. Conclusions and Recommendations

In this study, RS and GIS data and tools were effectively used in the assessment and estimation of the annual average soil loss in the Koromi-Federe catchment of Jos East LGA of Nigeria. The result showed that the influence of factors such as the soil type (*K*-factor), slope length/slope steepness (*LS*-factor), and rainfall (*R*-factor) exacerbated soil loss in the catchment area. Factors such as vegetated cover (*C*-factor) and management practices (*P*-factor) however reduced soil loss, but vegetation cover was not very effective in completely eradicating soil loss in hilly areas with steep slopes and under heavy rainfall. These hilly regions are famous hiking paths and grazing areas, this portrays dangers to cattle and human life if not properly managed. Our findings support previous reports that vegetation and precipitation suggest competing effects, implying the force of intense precipitation can reduce the impact of vegetation cover. We, therefore, conclude that erosion and soil loss in the Koromi–Federe catchment is aggravated by anthropogenic factors of improper farming practices and overgrazing like in most parts of Nigeria, but more influenced by the increase in rainfall and suitable environmental factors (steep slopes and soil). This is a key finding as it can guide sustainability practices to control erosion in the region. We, therefore, recommend sustainable cropping systems such as terracing, contour ridging, and effective cattle ranching for effective soil loss control in Jos East LGA and Nigeria. The Koromi–Federe catchment is located within hills and forms major river tributaries. This area will continue to encounter devastating soil loss especially now with increased rainfall intensity and duration reported in the region under climate change. The need for sustainable land management practices cannot be over-emphasized, and we recommend further studies to adequately understand vegetation–precipitation–soil loss relationships for effective and practical erosion control.

**Author Contributions:** Conceptualization, A.A.U., J.O.A. and E.S.I.; methodology, A.A.U., J.O.A. and E.S.I.; software, A.A.U., J.O.A. and E.S.I.; validation, A.A.U., J.O.A. and A.I.; formal analysis, A.A.U., J.O.A. and E.S.I.; investigation, E.N.G., E.S.I. and H.A.S.; resources, E.S.I.; data curation, A.A.U. and J.O.A.; writing—original draft preparation, A.A.U. and J.O.A.; writing—review and editing, A.A.U., J.O.A., A.I., E.N.G.; E.S.I. and H.A.S.; visualization, A.A.U., J.O.A. and E.S.I.; supervision, E.N.G., E.S.I. and H.A.S.; project administration, A.A.U., J.O.A. and A.I. All authors have read and agreed to the published version of the manuscript.

**Funding:** This research received no external funding.

**Acknowledgments:** We acknowledge the support National Centre for Remote Sensing Jos, which is an aegis of National Space Research Agency, Nigeria. We also acknowledge the state holders of the Jos East LGA under the leadership of Hon. Dauda Barde for giving us the opportunity to present our research findings, with a promise for implementation in due time to help farmers and good people of Jos East LGA how to curtail erosion expansion and boost sustainable food and cattle production in Jos East LGA.

**Conflicts of Interest:** The authors declare no conflict of interest.

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
