# Peer review of "Soil Loss Estimation Using Remote Sensing and RUSLE Model in Koromi-Federe Catchment Area of Jos-East LGA, Plateau State, Nigeria"

_2673-7418, doi:10.3390/geomatics2040027_

Round 1
Reviewer 1 Report
see attached file

Author Response
Thank you for your very insightful feedback, it has significantly improved the article. Please find attached details of the responses attached and the revisions in the manuscript

Reviewer 2 Report
Comments are attached!

Author Response
Thank you for your feedback, it has significantly improved the article. Please find attached details of the responses attached and the revisions in the manuscript

Reviewer 3 Report
The article is interesting and deals with the important topic of soil erosion. Although, it is now well referenced. Please provide more relative references to highlight the importance of the examined topic and the best author knowledge. The main concerns are listed below:
Line 43. Add a comment and relevant references: Thus, soil erosion is a major threat to biodiversity (Köninger et al., 2022)
Köninger, J., Panagos, P., Jones, A., Briones, M. J. I., & Orgiazzi, A. (2022). In defence of soil biodiversity: Towards an inclusive protection in the European Union. Biological Conservation, 268, 109475.
The legend in figure 1 is not very clear. Please improve the quality and increase the font height.
Line 75. The always-growing availability of Earth Observation data and the well-established use of Geographic Information Systems (GIS) lead to the development of automated geospatial toolboxes for the estimation of soil erosion (Stefanidis et al., 2021; Ahmari et al., 2022)
Stefanidis, S., Chatzichristaki, C., & Stefanidis, P. (2021). An ArcGIS toolbox for estimation and mapping soil erosion. J. Environ. Prot. Ecol, 22, 689-696.
Ahmari, H., Pebworth, M., Baharvand, S., Kandel, S., & Yu, X. (2022). Development of an ArcGIS-Pro Toolkit for Assessing the Effects of Bridge Construction on Overland Soil Erosion. Land, 11(9), 1586.
Line 87. Please provide some experiments about model accuracy from the literature from comparison with field measurements.
Line 89. Clearly state the novelty points of this approach and the research gap, answered from this study.
Line 240. Please provide a summary table with the data sources, format type, spatial resolution and availability link of the data used in the current approach.
The discussion section is rather small. Please discuss your result in a wider context and give some direction for future research.
Author Response
Thank you for your very insightful suggestions. Please find attached our responses and please see the suggested revisions in the manuscript.

Round 2
Reviewer 1 Report
Thank you for significant improvement of your manuscript.
Reviewer 2 Report
I congratulate the authors for addressing all the comments adequately and in well manner and recommend for acceptance.
Reviewer 3 Report
The author addressed all the reviewer comments